# Earnings Management and Status of Corporate Governance under Different Levels of Corruption—An Empirical Analysis in European Countries

**Ioannis Dokas** 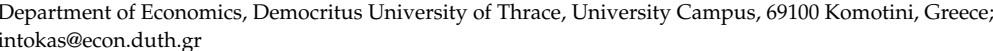

Department of Economics, Democritus University of Thrace, University Campus, 69100 Komotini, Greece; intokas@econ.duth.gr

**Abstract:** This study investigates the effect of the characteristics of the board of directors on the accrual and real earnings management level, focusing on the role of the corruption level. The employed dataset consists of 469 European-listed firms from 2011 to 2019. Using a fixed-effect panel data regression model, the results documented that larger boards lack coordination and communication in less corrupt economies, facilitating earnings manipulation through accruals and sales. In highly corrupt countries, oversized boards are associated with increased manipulation of production costs and discretionary expenses. Board meetings are positively related to accrual and sales manipulation in low-corruption countries, and board independence leads to reducing discretionary expenses regardless of corruption level. Board tenure negatively affects accruals and discretionary expenses but tends to increase manipulation through production costs in low-corruption contexts. Additionally, when the CEO serves as the board chairman, it encourages the manipulation of discretionary expenses while reducing real earnings manipulation through sales and production costs. In aggregate, the level of corruption can influence a board's effectiveness under specific conditions.

**Keywords:** earnings management; accrual earnings management; real earnings management; board characteristics; corporate governance; corruption

## 1. Introduction

The board of directors theoretically acts on behalf of stockholders, preserving the firm's value by monitoring, evaluating, and advising the managerial executives. As a part of their duties, the directors are charged with preventing earnings management practices that derive from the opportunistic behaviors of the managers. The board's role is vital to safeguard the credibility of the financial reporting procedure and avoid the adverse consequences of revealing low-quality financial information that misleads the stakeholders, especially in cases of listed firms. However, external social, legislative, and economic factors can affect the board's function and its effectiveness in monitoring the management. For instance, the level of corruption can reflect the efficacy of a political and legal national system, which can influence the behavior of board members and their potential to address corporate agency problems. Donadelli et al. (2014) documented that the level of corruption generates different behaviors of the board, and their characteristics present various effects on mitigating agency problems and preserving the firm's performance.

The growing earnings management literature includes a considerable number of studies investigating the direct relationship between the quality of financial reporting and the status of corporate governance, such as the characteristics of the board of directors. The quality of financial reporting is partially determined by the level of earnings management which occurs when managers intervene in the financial reporting procedure to modify the financial statements to achieve a specific goal that depends on the profitability (Healy and Wahlen 1999). Managers can engage in earnings management through accounting policies, known as accrual earnings management, or through adjustments on the operating

activities (revenue, expenses, and production cost), known as real earnings management (Kothari et al. 2016). Many board characteristics have previously been examined as determinants of the level of earnings management. The diversity of the knowledge and the experience of an oversized board has been reported to lead to lower earnings management levels (Ebrahim 2007; Al-Bassam et al. 2018; Saona et al. 2020; Cho and Chung 2022). On the contrary, the size of the board is associated with communication and collaboration problems that facilitate the opportunistic practices of managers (Lipton and Lorsch 1992; Jensen 1993; Jaggi and Leung 2007; Kapoor and Goel 2017; Abdou et al. 2021; Githaiga et al. 2022). The annual number of board meetings can indicate an active board that exerts effective monitoring duties improving the financial reporting quality (Lipton and Lorsch 1992; Jensen 1993; Vafeas 1999; Xie et al. 2003; Kang and Kim 2012; Attia et al. 2022; Usman et al. 2022). However, another part of the literature has questioned the linkage between earnings management and board meetings (Ebrahim 2007; Kjærland et al. 2020). Additionally, many studies have shown that the presence of independent directors enhances the monitoring ability of the board, limiting accounting (Kapoor and Goel 2017; Abdou et al. 2021; Hasan et al. 2022; Khan et al. 2022) and real activities manipulations (Kang and Kim 2012; Chen et al. 2015). Other studies did not find significant results that can support that the independent members contribute to the mitigation of earnings manipulations (Vafeas 2005; Agrawal and Chadha 2005; Larcker et al. 2007; Kjærland et al. 2020). Boards consisting of long-tenured members can offer more experience and knowledge to mitigate earnings management (Beasley 1996; El Diri et al. 2020; Usman et al. 2022) but can also lead to low-quality monitoring efficacy due to the long-term collaboration with the managerial executives (Vafeas 2005). Board members with industry-related and financial expertise detect and react to accounting manipulations and deviations of the operating activities, recognizing the negative consequences on the firm's value (Xie et al. 2003; Chen and Komal 2018; Githaiga et al. 2022; Zalata et al. 2022). When the CEO holds the position of the chairman, it offers these executives a gathered power that can increase earnings management due to opportunistic practices (Chang and Sun 2009; Saona et al. 2020; Usman et al. 2022; Alves 2023). Nevertheless, according to Yasser and Mamun (2015), many factors determine whether the separation of the positions of CEO and chairman is efficient in limiting the level of earnings manipulations, while other studies support that a CEO-chairman preserves the shareholders' interests by moderating earnings management (Gavana et al. 2022; Attia et al. 2022).

The board's behavior is susceptible to the political, social, and legislative conditions of the environment in which it operates. These external economic factors significantly determine the motivations for accrual and real earnings management (Ahmadi et al. 2023). A proxy for such factors is the corruption level that has been reported, which can influence the impact of board characteristics on the firm's performance (Donadelli et al. 2014). Moreover, several studies have documented the direct correlation between financial reporting quality and corruption (Lourenço et al. 2018; Xu et al. 2019; Christopoulos et al. 2022). However, the existing literature has yet to explore how corruption influences firms in addressing their agency problems and curbing managerial opportunism.

Most studies attempt to identify the ability of the board to detect and react to managers' manipulative accounting practices. However, three theories render real earnings management worth taking into consideration: firstly, the fact that firms determine the mix of their earnings management methods considering internal and external factors (Zang 2012; Christopoulos et al. 2022; Viana et al. 2023); secondly, real earnings management methods are deemed to be less detectable than accrual-based manipulations (Graham et al. 2005; Gunny 2010), because they require a higher level of industry-related expertise; and thirdly, real earnings management is associated with future financial underperformance due to sales manipulations and interventions on innovative activities such as R&D expenses (Vorst 2016; Bereskin et al. 2018). Under this assumption, board directors can observe the deviations from the usual operating activities and react to limit them, preventing negative financial consequences.

The relevant literature has provided evidence for the impact of board characteristics in samples of specific countries in Europe (Saona et al. 2020; Kjærland et al. 2020; Gavana et al. 2022) and worldwide (Rajeevan and Ajward 2020; Attia et al. 2022; Hasan et al. 2022; Cho and Chung 2022). Nevertheless, the literature has not yet focused on examining international samples to identify whether the existing findings are valid for diversified environments. The distinct and diversified European corporate landscape creates a conducive environment for conducting international-focused research. European firms are generally characterized by concentrated ownership primarily controlled by large single shareholders (e.g., investing groups and families). However, UK firms' ownership is more dispersed, influenced by USA characteristics (Bekiris and Doukakis 2011). Moreover, the legal systems that supervise corporate relationships and the function of stock markets are diversified but can be categorized into groups with similar principles. Regarding the corruption level, which is of great importance for this study, it varies across European firms but simultaneously is maintained at higher levels than in emerging economies because of the political and legal stability.

Overall, this study investigates which board characteristics constitute determinants of the financial reporting quality, paying particular attention to accrual and real earnings management methods. Further, considering the discrepancies in corruption levels across European countries, the research explores whether these differences in the political and legal environment influence the impacts of board characteristics on earnings management.

This research offers several contributions to the existing body of literature, examining which and to what extent board characteristics facilitate or deter managers from engaging in real and accrual-based earnings management policies, investigating whether the real earnings management theories can explain the phenomenon. Firstly, using a recent dataset, this research provides new evidence on the existing literature, which has exhibited mixed results for specific corporate governance variables. Secondly, the prior literature has paid particular attention to accrual-based earnings management methods. This study attempts to enrich the part of the literature that employs both accrual and real activity proxies to capture a more comprehensive range of manipulative behaviors. Thirdly, few studies have employed European samples, while the majority of them focused on specific-country firms. Using an inter-European dataset, this study broadens the range of the findings beyond the limitations of specific-country research. Lastly, this study introduces a novel dimension of the prominent research topic, investigating whether the impact of board characteristics on earnings management is diversified under different national corruption levels. In other words, the research examines to what extent the corruption level influences the dynamics of the board of directors' effectiveness.

The study findings reveal that larger boards in less corrupt environments often lack coordination and communication, allowing earnings manipulation through accruals, discretionary expenses, and sales. In highly corrupt nations, oversized boards are positively linked only to manipulating production costs and discretionary expenses. In low-corruption countries, more board meetings are associated with higher levels of accrual and sales manipulation, while board independence tends to decrease discretionary expenses, regardless of the corruption level. Additionally, board tenure has an adverse impact on accruals and discretionary expenses but tends to raise manipulation through production costs in low-corruption contexts. Notably, when the CEO also holds the position of board chairman, it encourages the manipulation of discretionary expenses while reducing real earnings manipulation through sales and production costs.

The content of this research is organized into three sections. Section 2 constitutes an overview of the past related literature. Section 3 elaborates on the methodology, including the sample features, the presentation of the research approach, and the variable selection process. Section 4 presents the tables of the results and their discussion concerning the key research outcomes and whether these findings align with or deviate from the existing literature.

## 2. Brief Literature Review

The agency problem derives from the conflict of interest between stockholders and managers (Fama and Jensen 1983). Earnings management is a corporate strategy employed by managers to meet or exceed specific financial targets aligning with shareholders' benefits or their interests (Degeorge et al. 1999).

The board of directors plays a crucial role in addressing the agency problem and its impact on earnings management (Jensen and Meckling 1976). The board is delegated to oversee the management's actions and represent the interests of shareholders. An effective board can reduce agency costs by properly monitoring, restricting managerial incentives, and implementing strict corporate governance mechanisms. Board characteristics can function as significant determinants in the degree of efficiency of the board and consequently can impact the level of earnings management (Beasley 1996; Dechow et al. 1996). The related literature has demonstrated the association of a series of board characteristics with the intensity of accrual or real earnings management under regular or special corporate conditions. Previous research primarily focuses on examining the impact of the board on managerial behavior by analyzing accrual earnings management (Xie et al. 2003; Ebrahim 2007; Jaggi and Leung 2007; Chen et al. 2015; Kapoor and Goel 2017; Abdou et al. 2021; Githaiga et al. 2022; Hasan et al. 2022). However, three points justify why real earnings management should be considered in the research context and to what extent the board members can detect such practices and react accordingly. The first reason stems from the findings of several studies that opt for accrual and real earnings management, considering various internal and external factors (Zang 2012; Christopoulos et al. 2022; Viana et al. 2023). The second is observed in the theory that real earnings management constitutes a less detectable method of earnings management (Graham et al. 2005; Gunny 2010; Badertscher 2011; Chi et al. 2011; Huang et al. 2021; Mughal et al. 2021; Christopoulos et al. 2022). In other words, managers can trade off between accrual and real earnings management, weighing the benefits of engaging in manipulations and the cost of revealing. The third reason is that real earnings management can result in problems in future performance, productivity, and competitive position (Vorst 2016; Bereskin et al. 2018).

The efficiency of a board is a crucial determinant of the firm's performance, profitability, low cost of equity, and overall value (Yermack 1996; Eisenberg et al. 1998; Fuerst and Kang 2004; Salehi et al. 2023). Numerous studies have demonstrated findings about the impact of board characteristics on the accrual and real earnings management policies. Dechow et al. (1996) suggested that when governance monitoring by the board is weak, managers are more inclined to manipulate earnings for their gain, highlighting the influence of board size on earnings management. A group of studies has extensively discussed the effect of the number of board members as a determinant of the level of earnings management, paying attention to the role of the quality of communication among the members. An oversized board lacks effective communication and coordination, which dilutes the board's function and results in restricted manager oversight (Lipton and Lorsch 1992; Jensen 1993). Likewise, Jaggi and Leung (2007), Kapoor and Goel (2017), Abdou et al. (2021), Githaiga et al. (2022), and Hasan et al. (2022) confirmed a positive association between board size and discretionary accruals. These studies imply that larger boards may be more prone to allowing managers to engage in manipulation through accounting practices. On the contrary, other studies, such as those conducted by Xie et al. (2003), Bradbury et al. (2006), Ebrahim (2007), Al-Bassam et al. (2018), Saona et al. (2020), and Cho and Chung (2022), have demonstrated an inverse association between board size and discretionary accruals. This relationship is explained by the fact that oversized boards, including more experienced and qualified members, can actively intervene in accounting procedures to control earnings manipulation. Regarding real earnings management, Kang and Kim (2012) found that the board size constitutes a restricting factor of manipulation through production cost, while Rajeevan and Ajward (2020) did not report a significant relationship between real earnings management and board size.

Studies like Lipton and Lorsch (1992), Jensen (1993), and Vafeas (1999) established the notion that the number of board meetings is an indicator of an active and efficient board that systematically monitors and controls the management decisions preserving the shareholders' interests. Similarly, other studies conducted by Xie et al. (2003), Usman et al. (2022), and Hasan et al. (2022) presented a negative correlation between discretionary accrual and the number of meetings, illustrating that the frequency of board meetings enhances the board's power and improves the quality of accruals. On the other hand, Ebrahim (2007) and recently Kjærland et al. (2020) did not confirm a significant correlation between board frequency and accrual earnings management proxies. Concerning the relationship between board meetings and real earnings management, Attia et al. (2022) exhibited that a high frequency of meetings facilitates managerial discretion on production cost while not observing significant results for manipulations through discretionary expenses. Also, Attia et al. (2022), similar to Kang and Kim (2012), found that board meetings are negatively related to the level of manipulation through sales.

Another board feature that has been the subject of research is the number of independent members. The literature exhibits mixed findings about the impact of board independence on earnings management, which can be attributed to endogeneity concerning the corporate governance variables (Larcker et al. 2007). Prior studies (Klein 2002; Xie et al. 2003; Be' dard et al. 2004; Ebrahim 2007; Chen et al. 2015; Busirin et al. 2015; Kapoor and Goel 2017; Abdou et al. 2021; Hasan et al. 2022; Khan et al. 2022) have established the negative effect of board independence on the discretionary accruals. In essence, external and independent members contribute to the efficient supervision and monitoring of the management activities related to the financial reporting process. However, other studies (Vafeas 2005; Agrawal and Chadha 2005; Larcker et al. 2007; Kjærland et al. 2020) did not observe a specific significant association between accrual-based manipulations and board independence indices. Little attention has been paid to real earnings management, providing mixed findings. Kang and Kim (2012) found that an aggregate variable of real earnings management decreases as board independence increases. However, in a more thorough analysis, the results indicated the opposite direction for the effect of board independence on the level of sales manipulation. As a part of their research, Chen et al. (2015) examined the effect of board independence on an aggregate earnings management proxy combining manipulation methods through production cost and discretionary expenses. Similar to Kang and Kim (2012), the results indicated that increases in board independence deter managers from engaging in real earnings management, but without defining the specific impact of the two manipulation methods. More recently, Attia et al. (2022) found a positive association between board independence and abnormal levels of discretionary expenses, while their results indicated a negative relationship between board independence and abnormal production cost and cash flows (sales manipulations).

The board tenure can signify the shareholders' trust and satisfaction with the board's monitoring abilities (Livnat et al. 2021). A longer-tenure board theoretically consists of more experienced members familiar with the corporate procedures, providing efficient directions to managers and preserving the firm's value. According to Beasley (1996), extended tenure of board members can reduce the probability of the company engaging in fraudulent activities. El Diri et al. (2020) revealed that in low-concentrated markets, there is a negative association between board tenure and the level of earnings management. However, in concentrated markets, firms with board members serving long periods tend to employ real earnings management strategies instead of accrual-based methods, which are considered less easily detectable. Usman et al. (2022) found that board tenure is negatively related to discretionary accruals, attributing this finding to the experience and expertise in monitoring managers' activities. Other studies, like Dhaliwal et al. (2010) and Ghosh et al. (2010), confirmed the negative association between tenure and earnings management, focusing on audit committee characteristics. Additionally, Park and Shin (2004) and Sun et al. (2014) found no significant evidence that board tenure mitigates earning manipulation. Another group of studies (Xie et al. 2003; Vafeas 2005) presented that long board tenure leads to

lower accrual quality, which shows that directors who have served on the board for a more extended period might be less efficient in overseeing and monitoring the company's activities, potentially due to becoming too aligned with management.

The existing literature has also focused on the significance of qualified directors with specific industry knowledge or expertise in finance and accounting to maintain the quality of financial reporting (Mousavi et al. 2022). When board members have a strong understanding of financial reporting procedures, they can monitor managers effectively, provide guidance, and influence other board members (Güner et al. 2008; Jeanjean and Stolowy 2009; Mousavi et al. 2022). Several studies provided evidence that the financial expertise of board members or audit committee directors contributes to a low level of accrual-based manipulations (Nelson and Devi 2013; Badolato et al. 2014; Chen and Komal 2018; Githaiga et al. 2022; Zalata et al. 2022). Nevertheless, recently, Le and Nguyen (2023), found no significant impact of financial expertise on the level accrual-based manipulations. Regarding the effect of specific industry qualifications, Xie et al. (2003) demonstrated that corporate experience significantly mitigates accounting manipulations. Additionally, industry-expert directors intervene in the decision-making process related to operational activities like R&D expenses, which constitutes the usual real earnings management method (Faleye et al. 2018). However, (Wang et al. (2015) found no connection between real earnings management and the presence of industry-expert independent directors.

The presence of CEO and chairman duality is a typical corporate governance structure, and its impact on earnings management has been extensively discussed in the literature. Jensen (1993) and Lipton and Lorsch (1992) supported the opinion that separating the roles of CEO and chairman is undeniably more efficient for the essential tasks of the board, as it helps mitigate earnings management tactics that primarily benefit the CEO personally. Furthermore, CEO–chairman duality signals to markets a limited ability of the board to mitigate opportunistic manipulation activities (Gul and Lai 2002; Anderson et al. 2003). Numerous studies (Chang and Sun 2009; Saona et al. 2020; Usman et al. 2022; Alves 2023) have confirmed that CEO duality allows SEOs to engage in earnings management more intensively affecting negatively the accrual quality and the informativeness of the financial reporting. However, Yasser and Mamun (2015) supported the idea that various factors can influence the association between earnings quality and CEO duality, and the separation of the two positions is not always appropriate for improving the monitoring process. In addition, other studies (Xie et al. 2003; Bradbury et al. 2006; Ebrahim 2007; Al-Haddad and Whittington 2019; Le and Nguyen 2023) found no significant results to support a specific impact of CEO duality and accrual-based manipulations. In terms of real earnings management, Nuanpradit (2019) reported that CEO duality leads to a higher level of sales manipulation. On the contrary, Gavana et al. (2022) and Attia et al. (2022) suggested that the CEO, who serves as board chair, preserves the shareholders' interests by mitigating the real earnings management practices through sales and discretionary expenses. Al-Haddad and Whittington (2019), using a combination of real earnings management proxies, provided evidence that, in aggregate, CEO duality leads to higher levels of real earnings management. However, the results generated by models using proxies for specific manipulation methods appeared weaker.

Undoubtedly, the past literature has demonstrated a substantial body of research outcomes manifesting the impact of board characteristics on the exacerbation of agency problems through earnings management. The board's quality and behavior could be influenced by political, social, and legislative conditions of the economic environment in which the firm operates. The national level of corruption can be a proxy incorporating such information for the quality of a country's political and legal system. Studies such as Donadelli et al. (2014) confirmed that the influence of board characteristics on firm performance varies across industries with different levels of sensitivity to corruption. On the other hand, the external economic environment that a firm operates in can significantly influence the motivation and the level of accrual and real earnings management (Ahmadi et al. 2023). Many studies have also shown the direct association between the quality of

financial reporting and corruption, indicating that management behavior can result from the efficacy of the national legal system (Lourenço et al. 2018; Xu et al. 2019; Christopoulos et al. 2022). However, these studies do not focus on the extensions of the agency problem and the conflict between ownership and management that leads to earnings management. Overall, the existing body of literature has not delved into whether corruption plays a role in shaping how firms address their agency problems, mitigating the opportunistic practices of the managers.

## 3. Steps of Methodology—Variables Selection

### 3.1. Sample Selection

This research employs a dataset including 3292 observations of firm-year data from 469 European firms during the period between 2011 and 2019. The examined period is limited to 2019 to avoid the effect of the COVID-19 pandemic. The earnings management motivations are susceptible to changes in economic cycles, particularly in the presence of unforeseen events (Yan et al. 2022). The sample includes industrial and commercial enterprises, with at least five firms per industry. Similar to past research (Armstrong et al. 2016; Haga et al. 2018; Chang and Pan 2020; Christensen et al. 2022; Al-Shattarat et al. 2022), firms from the financial sector are excluded due to the challenges associated with estimating accrual and real activity earnings management measures. The industries and countries in which the sample firms operate are presented in Table 1. Additionally, to ensure compliance with common accounting standards and minimize missing data, the firms included in the study are listed in European stock exchange markets having adopted IAS-IFRS. The financial and corporate governance information was obtained from the Eikon Thomson-Reuters platform and, in some cases, from the firms' annual reports.

**Table 1.** Industry and nation frequency table.

| Industry Group Name | Freq. | Country of Headquarters | Freq. |
| --- | --- | --- | --- |
| Computers, Phones, and Household Electronics | 5 | Lithuania | 1 |
| Personal and Household Products and Services | 5 | Slovenia | 1 |
| Construction Materials | 6 | Croatia | 1 |
| Electronic Equipment and Parts | 6 | Iceland | 1 |
| Healthcare Providers and Services | 6 | Estonia | 2 |
| Passenger Transportation Services | 6 | Cyprus | 2 |
| Containers and Packaging | 7 | Luxembourg | 4 |
| Food and Drug Retailing | 8 | Greece | 5 |
| Beverages | 9 | Portugal | 5 |
| Household Goods | 9 | Belgium | 10 |
| Paper and Forest Products | 9 | Austria | 12 |
| Specialty Retailers | 11 | Ireland; Republic of | 13 |
| Semiconductors and Semiconductor Equipment | 15 | Norway | 14 |
| Textiles and Apparel | 15 | Netherlands | 17 |
| Biotechnology and Medical Research | 16 | Italy | 18 |
| Freight and Logistics Services | 16 | Spain | 19 |
| Healthcare Equipment and Supplies | 19 | Denmark | 21 |
| Homebuilding and Construction Supplies | 21 | Finland | 26 |
| Automobiles and Auto Parts | 23 | France | 45 |
| Construction and Engineering | 27 | Sweden | 47 |
| Food and Tobacco | 27 | Switzerland | 47 |
| Metals and Mining | 28 | United Kingdom | 57 |
| Pharmaceuticals | 30 | Germany | 101 |
| Chemicals | 36 | | |
| Software and IT Services | 36 | | |
| Machinery, Tools, Heavy Vehicles, Trains, and Ships | 73 | | |
| Total | 469 | Total | 469 |

*3.2. Earnings Management Estimation*

The initial stage of the empirical analysis focuses on estimating earnings management to quantify the degree of managerial discretion over accounting figures and operational activities. The related literature has provided numerous approaches associated with the estimation of accrual earnings management. The prevailing "family" of models is based on the classic Jones (1991) model, which assumes that total accruals can be divided into discretionary (abnormal) and non-discretionary (normal) accruals. The discretionary portion of total accruals is the accounts that managers can modify to achieve their financial targets, while non-discretionary accruals are the accounts that represent the firm's financial performance and cannot be used as a manipulation instrument (Healy 1985). This research employs the McNichols (2002) model, which constitutes a modified version of the classic Jones (1991) model combined with the approach suggested by Dechow and Dichev (2002). By implementing linear regression on the following equation, the generated residuals are used as the discretionary component of total accruals, the proxy for earnings management.

$$\frac{TA_{it}}{A_{it-1}} = b_0 + b_1\left(\frac{CFO_{it-1}}{A_{it-1}}\right) + b_2\left(\frac{CFO_{it}}{A_{it-1}}\right) + b_3\left(\frac{CFO_{it+1}}{A_{it-1}}\right) + b_4\left(\frac{\Delta sales_{it}}{A_{it-1}}\right) + b_4\left(\frac{PPE_{it}}{A_{it-1}}\right) + e_{it} \tag{1}$$

Several past studies have used the McNichols (2002) model (Stubben 2010; Yasser and Mamun 2015; Ali and Zhang 2015; Koo et al. 2017; Capalbo et al. 2018; Ham et al. 2017; El Diri et al. 2020). Its fundamental assumption is that the current earnings reflect the accruals and current cash flows from operating activities and serve as an indicator for projecting future cash flows. Total accruals (*TA*), which constitute the dependent variable, are computed as net income before extraordinary items minus cash flows from operations. The independent variables are the cash flows from operations in the previous period ($CFO_{t-1}$), in current period ($CFO_t$), and in the following period ($CFO_{t+1}$). Also, the change in total sales ($\Delta sales$) is included to control firms' financial performance and is assumed to be a non-discretionary item, and the gross property plant and equipment (*PPE*) account represents the non-discretionary part of depreciations. All variables are scaled by lagged total assets to reduce heteroscedasticity.

Consistent with the majority of prior studies on real earnings management (Enomoto et al. 2015; Anagnostopoulou and Tsekrekos 2017; Oz and Yelkenci 2018; Pappas et al. 2019; Chang and Pan 2020; Elrazaz et al. 2021; Mughal et al. 2021), this research adopts the Roychowdhury (2006) models to measure the extent of manipulation derived from operational activities, including methods through sales, discretionary expenses, and overproduction.

The first model among the three posits that temporary discounts and relaxed credit terms can boost both total sales and net income but, at the same time, lead to abnormally declined cash flows from operations. The following model defines the relationship between total cash flows from operations (*CFO*) and current sales (*St*) as well as the change in sales ($\Delta St$). Through linear regression, the obtained residuals represent the abnormal *CFO* or the proxy for sales manipulations.

$$\frac{CFO_{it}}{A_{it-1}} = a_0 + a_1\left(\frac{1}{A_{it-1}}\right) + \beta_1\left(\frac{S_{it}}{A_{it-1}}\right) + \beta_2\left(\frac{\Delta S_{it}}{A_{it-1}}\right) + \varepsilon_{it} \tag{2}$$

Additionally, reducing expenses, such as research and development (R&D) and selling, general, and administrative expenses (SG&A), can inflate reported profitability artificially. The following regression model estimates the abnormal portion of these discretionary expenses (R&D and SG&A), represented by the residuals.

$$\frac{DISEXP_{it}}{A_{it-1}} = a_0 + a_1\left(\frac{1}{A_{it-1}}\right) + \beta\left(\frac{S_{it}}{A_{it-1}}\right) + \varepsilon_{it} \tag{3}$$

Producing abnormally large quantities of products can function as a method of real earnings management. Increasing production can boost reported earnings by decreasing the cost of goods sold (*COGS*). Specifically, when firms engage in upward earnings man-

agement, the abnormal production cost tends to increase. The total production cost is the sum of *COGS* and the change in inventories (ΔINV). The subsequent model defines total costs based on current sales (*St*), the change in sales (ΔSt), and the lagged change in sales (ΔS_{t−1}). The residuals obtained from the linear regression represent the abnormal portion of the production cost, which equals the level of earnings management attributed to the overproduction.

$$\frac{PROD_{it}}{A_{it-1}} = a_0 + a_1 \left(\frac{1}{A_{it-1}}\right) + \beta_1 \left(\frac{S_{it}}{A_{it-1}}\right) + \beta_2 \left(\frac{\Delta S_{it}}{A_{it-1}}\right) + \beta_2 \left(\frac{\Delta S_{it-1}}{A_{it-1}}\right) + \varepsilon_{it} \quad (4)$$

In aggregate, four proxies for earnings management are generated to capture the level of management discretion on the accruals and operating activities. All models are implemented cross-sectionally by industry and year.

*3.3. The Empirical Approach*

Once the proxies for earnings management have been estimated, the next stage of the empirical analysis involves the examination of the impact of board characteristics on the level of earnings management. This is accomplished through the use of the following panel regression model (See Appendix A for definition of variables):

$$|EM_{i,t}| = a_o + a_1(BSize_{i,t}) + a_2(Meet_{i,t}) + a_3\left(Indep_{i,t}\right) + a_4(Tenure_{i,t}) + a_5(Skills_{i,t}) + a_6(Dual_{i,t}) + a_7(BIG4_{i,t}) +$$
$$a_8(ROA_{i,t}) + a_9(Fsize_{i,t}) + a_{10}(LEV_{i,t}) + Year\_Dummies + Country\_Dummies + Industry\_Dummies + \varepsilon_{i,t} \quad (5)$$

The model's dependent variable is the level of earnings management, represented by the four alternative proxies (|EMi,t|). The four alternative dependent variables are the discretionary accruals (*DACC*), the abnormal level of discretionary expenses (*abnDiscExp*), the abnormal production cost (*abnProd*), and the abnormal cash flows from operations (*abnCFO*). The sign of the proxies of earnings management indicates whether the earnings management practices are used for income-increasing (positive sign) or income-decreasing (negative sign) purposes. However, this research focuses on the impact of board characteristics on the intensity (level) of earnings management, regardless of the direction (income increasing/decreasing); hence, the proxies of earnings management are employed on their absolute value, as suggested by other past studies (Enomoto et al. 2015; Chowdhury et al. 2018; Haga et al. 2018; Tahir et al. 2019; Srivastava 2019; Li 2019; Jamadar et al. 2022; Al-Shattarat et al. 2022). The model includes four independent variables representing the board of directors' characteristics. Particularly, where "*Bsize*" is the board size defined as the natural logarithm of the number of the members of the board, "*Meet*" equals the natural logarithm of the number of annual board meetings, "*Indep*" is defined as the portion of independent members of the board, "*Tenure*" is the mean of the tenure of the board members, "*Skills*" is the portion of the board members that have acquired educational financial or industry-related skills, and the variable "*Dual*" is a dummy variable that equals one for firm years in which the CEO serves simultaneously as chair of the board.

Moreover, the proposed model contains firm-related financial control variables concerning financial performance, the scrutiny level, and earnings management incentives. In particular, the quality of auditors (*BIG4*) also plays a significant role in determining the quality of earnings. The involvement of Big 4 auditors can act as a deterrent to accrual-based manipulation because of their expertise and experience in detecting and reacting to earnings management practices (Becker et al. 1998; Francis et al. 1999; Francis and Wang 2008). The regression model also incorporates the return on assets (*ROA*) ratio as a control for firm performance. Firms in periods of profitability growth are motivated to employ earnings management techniques that raise their reported income. The firm size (*Fsize*) has a negative relationship with earnings management, as larger companies encounter higher scrutiny and external monitoring, making it costlier for them to be detected for manipulating financial statements (Watts and Zimmerman 1978). However, larger firms also undergo increased pressure to meet or exceed market expectations, leading to a greater motivation

to engage in earnings management to maintain their market performance (Barton and Simko 2002). The leverage ratio (*LEV*) reflects the incentives stemming from the need for earnings manipulation to prevent a breach of contract terms. Several studies have presented evidence suggesting that financially distressed firms are more driven to engage in earnings management to meet the contractual obligations imposed by their lenders (DeFond and Jiambalvo 1994; Sweeney 1994; Francis et al. 2005; Francis and Wang 2008; Anagnostopoulou and Tsekrekos 2017; Dyreng et al. 2020).

The model is implemented in panel data with industry (*Industry_Dummies*), nation (*Country_Dummies*), and year (*Year_Dummies*) fixed effects to consider the effect of factors that may vary across entities but remain constant over time. Also, the model is consistent with heteroscedasticity because it was applied with robust standard errors.

## 4. Empirical Results and Discussion

### 4.1. Main Results

As a preliminary step in the empirical analysis, Table 2 presents a correlation matrix to gain insights into the comprehensive linear relationships between variables. Moreover, the correlation matrix functions as a method to detect the presence of multicollinearity. Multicollinearity occurs when independent variables are highly correlated, leading to unreliable and unstable coefficient estimates. Table 2 shows that none of the pairs of independent variables exhibit a correlation exceeding the threshold of 0.8, indicating the absence of multicollinearity.

**Table 2.** Correlation matrix.

| Variables | (1) | (2) | (3) | (4) | (5) | (6) | (7) | (8) | (9) | (10) | (11) | (12) | (13) | (14) |
|---|---|---|---|---|---|---|---|---|---|---|---|---|---|---|
| (1) DACC | 1.000 | | | | | | | | | | | | | |
| (2) abnDiscExp | 0.073 * | 1.000 | | | | | | | | | | | | |
| (3) abnProd | 0.569 * | 0.186 * | 1.000 | | | | | | | | | | | |
| (4) abnCFO | 0.786 * | 0.078 * | 0.682 * | 1.000 | | | | | | | | | | |
| (5) Bsize | −0.089 * | −0.071 * | −0.049 * | −0.070 * | 1.000 | | | | | | | | | |
| (6) Meet | 0.042 | −0.044 * | −0.027 | 0.020 | −0.046 * | 1.000 | | | | | | | | |
| (7) Indep | 0.031 | −0.198 * | −0.101 * | −0.015 | −0.039 * | 0.024 | 1.000 | | | | | | | |
| (8) Tenure | −0.067 * | −0.037 | 0.008 | −0.014 | −0.018 | −0.145 * | −0.124 * | 1.000 | | | | | | |
| (9) Skills | 0.122 * | 0.018 | 0.017 | 0.070 * | −0.270 * | 0.076 * | 0.031 | 0.020 | 1.000 | | | | | |
| (10) Dual | −0.079 * | −0.003 | −0.051 * | −0.057 * | 0.125 * | −0.009 | −0.006 | 0.190 * | 0.129 * | 1.000 | | | | |
| (11) BIG4 | −0.052 * | 0.067 * | −0.005 | −0.035 | −0.016 | 0.037 | −0.064 * | 0.001 | −0.043 * | −0.039 * | 1.000 | | | |
| (12) ROA | 0.239 * | −0.047 * | 0.159 * | 0.243 * | −0.034 | −0.043 * | 0.034 | 0.031 | 0.004 | 0.080 * | | 1.000 | | |
| (13) Fsize | −0.201 * | −0.242 * | −0.107 * | −0.135 * | 0.520 * | 0.032 | 0.145 * | −0.006 | −0.123 * | 0.124 * | 0.099 * | 0.017 | 1.000 | |
| (14) LEV | 0.003 | −0.122 * | −0.042 * | −0.010 | 0.130 * | 0.096 * | 0.080 * | −0.065 * | 0.009 | 0.020 | −0.039 * | −0.235 * | 0.195 * | 1.000 |

* significant at less than 5% level.

Table 2 reveals that *Bsize* has a robust negative connection with all indicators of earnings manipulation. The number of board meetings (*Meet*) is notably linked to abnormal discretionary expenses. The proportion of independent members (*Indep*) correlates significantly with real earnings manipulation indicators such as *abnDiscExp* and *abnProd*. The average length of board tenure (*Tenure*) exhibits a negative association exclusively with discretionary accruals, whereas the variable skills demonstrate a negative relationship with *DACC* and *abnCFO*. The presence of CEO–chairman duality is negatively correlated with earnings management indicators, except for *abnDiscExp*.

Table 3 summarizes the outcomes obtained from the proposed model, which aims to examine the influence of corporate governance variables on the extent of accrual and real earnings management.

The findings of Table 3 suggest a strong positive relationship between board size (*Bsize*) and earnings management practices. Specifically, *Bsize* shows a significant association with discretionary accruals (*DACC*) and the real earnings management proxies *abnDiscExp* and *abnProd*. However, the association between *Bsize* and *abnCFO* is also positive but does not reach a significant level. These findings support previous studies (Kao and Chen 2004; Beiner et al. 2004; Jaggi and Leung 2007; Kapoor and Goel 2017; Abdou et al. 2021; Githaiga et al. 2022; Hasan et al. 2022) that suggest that larger boards may be less effective due to difficulties in coordination and communication, allowing management to manipulate accounting figures and operating activities. The regression results concerning *Bsize* contradict the results provided by the correlation matrix in Table 2. While providing valuable insights

into the bivariate relationships between variables, the correlation matrix may not fully capture the intricate relationships between multiple variables since the regression model incorporates control variables that may influence the relationship between the variables.

**Table 3.** Regression analysis results.

|  | (1) | (2) | (3) | (4) |
|---|---|---|---|---|
|  | DACC | abnDiscExp | abnProd | abnCFO |
| Bsize | 0.019 ** | 0.044 *** | 0.046 * | 0.014 |
|  | (0.008) | (0.012) | (0.024) | (0.011) |
| Meet | 0.01 * | −0.014 ** | −0.009 | 0.011 |
|  | (0.005) | (0.006) | (0.015) | (0.008) |
| Indep | 0 * | −0.001 *** | 0 ** | 0 |
|  | (0) | (0) | (0) | (0) |
| Tenure | −0.014 *** | −0.016 ** | 0.031 * | −0.002 |
|  | (0.005) | (0.007) | (0.016) | (0.005) |
| Skills | 0.006 * | 0.005 | −0.008 | 0.003 |
|  | (0.003) | (0.004) | (0.01) | (0.006) |
| Dual | 0 | 0.011 * | −0.024 * | −0.013 *** |
|  | (0.004) | (0.006) | (0.014) | (0.004) |
| BIG4 | −0.001 | 0.02 *** | 0.032 *** | −0.005 |
|  | (0.004) | (0.005) | (0.012) | (0.005) |
| ROA | 0.374 *** | −0.094 *** | 0.449 *** | 0.573 *** |
|  | (0.066) | (0.014) | (0.061) | (0.071) |
| Fsize | −0.009 *** | −0.015 *** | −0.023 *** | −0.006 ** |
|  | (0.003) | (0.002) | (0.005) | (0.003) |
| LEV | 0.03 ** | −0.081 *** | 0.02 | 0.016 |
|  | (0.015) | (0.015) | (0.033) | (0.023) |
| _cons | 0.129 *** | 0.284 *** | 0.37 *** | 0.056 |
|  | (0.047) | (0.046) | (0.097) | (0.044) |
| Observations | 1240 | 1822 | 1548 | 1858 |
| R-squared | 0.507 | 0.22 | 0.227 | 0.548 |

Robust standard errors are in parentheses. *** $p < 0.01$, ** $p < 0.05$, * $p < 0.1$.

The results for the impact of annual board meetings on earnings management are relatively weak since only the relationship with abnormal discretionary expenses is significant. Specifically, the variable Meet is significantly adversely related with only *abnDiscExp*, while its impact on *DACC* is significant only at 10%. In essence, when board meetings occur more frequently, managers engage in accrual earnings management, mitigating the use of real earnings management through discretionary expenses. This finding can be interpreted by the theory of the negative long-term consequences of real earnings management and primarily through discretionary expenses (Vorst 2016; Bereskin et al. 2018; Habib et al. 2022). In other words, active and vigilant boards deter managers from manipulating operating activities, prioritizing long-term value creation and the shareholders' benefits. The positive impact of board meetings on discretionary accruals is in line with Hasan et al. (2022), which supported that board meetings aim to confront corporate challenges and less so to monitor the management discretion on financial statements. Moreover, studies like Xie et al. (2003) and Usman et al. (2022) concluded the opposite findings, indicating that board meetings limit accounting manipulations. The adverse relationship between *abnDiscExp* and board meetings is not consistent with Attia et al. (2022), which demonstrated a positive impact of the frequency of board meetings on abnormal discretionary expenses.

Table 3 presents significant findings concerning the influence of board independence on abnormal discretionary expenses and production costs while yielding weaker results for models using discretionary accruals and abnormal *CFO* as dependent variables. Specifically, the variable *Indep* exhibits a negative association with abnormal discretionary expenses (*abnDiscExp*) and a positive association with abnormal production costs (*abnProd*). Essentially, boards composed of a higher proportion of external members discourage managers from

engaging in manipulative practices through discretionary expenses while simultaneously allowing them to intervene in production costs to modify the profitability. These findings contradict those of Attia et al. (2022), who demonstrated that board independence is positively associated with abnormal discretionary expenses and negatively associated with abnormal production costs and cash flows. The weak positive impact of board independence confirms the results of Kjærland et al. (2020) but are not consistent with numerous studies that support that external board members preserve the financial reporting quality (Xie et al. 2003; Ebrahim 2007; Chen et al. 2015; Klein 2002; Busirin et al. 2015; Hasan et al. 2022; Kapoor and Goel 2017; Abdou et al. 2021; Khan et al. 2022).

The negative relationship between board tenure and earnings management proxies of discretionary accruals and abnormal discretionary expenses suggests that longer-tenured board members have the expertise and knowledge to identify the adverse effects of earnings management. Studies like Forbes and Milliken (1999) and Usman et al. (2022) support the idea that the directors' knowledge constrains managers, discouraging them from manipulating financial reports. The statistical significance observed in models with the dependent variables *DACC* and *abnDiscExp* confirms that board tenure is associated with lower levels of earnings management through accruals and discretionary expenses. Additionally, board members with longer tenure are motivated to preserve their reputation and thus contribute to maintaining high-quality financial reports (Livnat et al. 2021).

The findings indicate a relatively limited impact regarding the effect of directors with specific skills on the earnings management proxies. Only the connection with discretionary accruals demonstrates statistical significance at a 10% level. The linkage of the variable *Skills* with *DACC* implies that directors with specific skills allow managers to engage in accounting manipulations. On the contrary, prior research (Xie et al. 2003; Githaiga et al. 2022) demonstrated that corporate or financial experts on a board are related to lower discretionary accruals. Regarding the real earnings management, similar to Khan et al. (2022), the results do not indicate any significant correlations to draw strong conclusions.

The presence of a CEO who also serves as the board's chairman appears to have contradictory impacts on earnings management. Like Bouaziz et al. (2020) and Alves (2023), the findings indicate that CEO duality is positively associated with the measure of manipulation through discretionary accruals, but not at a significant level. However, the effect of CEO duality on real earnings management proxies is presented as positive on the level of abnormal discretionary expenses and negative on the abnormal production cost, but both coefficients are not adequately significant. The impact of the variable *Dual* on the proxy of sales manipulation is negative at a high level of significance. This implies that when the roles of CEO and chairman are combined, earnings management is increased through discretionary expenses but acts as a deterrent to manipulating sales and production costs for managers. A possible explanation derives from the fact that experienced managers comprehend the detrimental effects of sales manipulation on the firm's future performance and value (Cohen and Zarowin 2010; Roychowdhury 2006; Huang and Sun 2017). Furthermore, board members care about their reputation (Srinivasan 2005; Badolato et al. 2014; Livnat et al. 2021) and tend to avoid jeopardizing their fame by employing such value-decreasing strategies. This finding contradicts Nuanpradit (2019) and Rajeevan and Ajward (2020), who demonstrated a positive association between CEO duality and sales manipulation level.

In more thorough research, Table 4 displays the outcomes of implementing the proposed model to groups of firm observations operating in highly and less corrupt economic environments. The corruption level is determined using the Corruption Perception Index (referred to as CPI), which is computed by the Transparency International Organization. The CPI assesses the extent of corruption in the national public sector, with scores ranging from 0 to 100. Nations with scores close to zero exhibit high levels of corruption, while those nearing 100 demonstrate low levels. The separation of the sample was achieved by using the milestone of the value 81, which equals the median CPI score across the entire employed sample.

**Table 4.** Regression model after separation of the entire sample into low corrupt and highly corrupt countries.

| Dependent Variable: | DACC | | abnDiscExp | | abnProd | | abnCFO | |
|---|---|---|---|---|---|---|---|---|
| | (1) High Corruption | (2) Low Corruption | (3) High Corruption | (4) Low Corruption | (5) High Corruption | (6) Low Corruption | (7) High Corruption | (8) Low Corruption |
| Bsize | 0.009 | 0.029 *** | 0.04 *** | 0.042 *** | 0.086 *** | 0.033 | −0.004 | 0.027 ** |
| | (0.01) | (0.01) | (0.013) | (0.015) | (0.032) | (0.034) | (0.02) | (0.011) |
| Meet | 0 | 0.019 *** | −0.009 | −0.003 | −0.017 | 0.006 | 0.009 | 0.014 ** |
| | (0.008) | (0.007) | (0.007) | (0.008) | (0.023) | (0.02) | (0.016) | (0.006) |
| Indep | 0 * | 0 | −0.001 *** | −0.001 *** | 0 | −0.001 ** | 0 | 0 |
| | (0) | (0) | (0) | (0) | (0) | (0) | (0) | (0) |
| Tenure | −0.018 *** | −0.01 * | −0.003 | −0.008 | 0.01 | 0.05 ** | 0.006 | −0.008 |
| | (0.006) | (0.006) | (0.008) | (0.01) | (0.027) | (0.022) | (0.008) | (0.006) |
| Skills | 0.007 | 0.003 | 0.008 * | 0.003 | −0.002 | −0.014 | −0.009 | 0.008 * |
| | (0.004) | (0.004) | (0.005) | (0.005) | (0.016) | (0.012) | (0.015) | (0.004) |
| Dual | 0 | −0.002 | −0.002 | 0.005 | −0.027 | −0.007 | −0.009 | −0.013 ** |
| | (0.006) | (0.006) | (0.007) | (0.008) | (0.026) | (0.015) | (0.007) | (0.006) |
| BIG4 | −0.005 | 0.001 | 0.028 *** | 0.027 *** | 0.053 *** | 0.017 | −0.01 | 0.001 |
| | (0.004) | (0.007) | (0.006) | (0.007) | (0.016) | (0.018) | (0.009) | (0.005) |
| ROA | 0.339 *** | 0.394 *** | −0.081 *** | −0.091 *** | 0.416 *** | 0.457 *** | 0.636 *** | 0.537 *** |
| | (0.078) | (0.083) | (0.014) | (0.019) | (0.111) | (0.078) | (0.097) | (0.101) |
| Fsize | −0.005 * | −0.012 *** | −0.015 *** | −0.015 *** | −0.05 *** | −0.011 * | −0.004 | −0.008 *** |
| | (0.003) | (0.004) | (0.002) | (0.003) | (0.008) | (0.006) | (0.004) | (0.003) |
| LEV | 0.017 | 0.044 ** | −0.063 *** | −0.042 * | 0.022 | 0.024 | −0.019 | 0.065 ** |
| | (0.019) | (0.022) | (0.019) | (0.022) | (0.042) | (0.049) | (0.037) | (0.026) |
| _cons | 0.09 * | 0.186 ** | 0.321 *** | 0.369 *** | 0.906 *** | 0.394 *** | 0.074 | 0.083 |
| | (0.05) | (0.08) | (0.048) | (0.061) | (0.155) | (0.117) | (0.057) | (0.059) |
| Observations | 446 | 794 | 1347 | 1033 | 626 | 922 | 808 | 1050 |
| R-squared | 0.598 | 0.498 | 0.208 | 0.215 | 0.324 | 0.201 | 0.533 | 0.599 |

Robust standard errors are in parentheses. *** $p < 0.01$, ** $p < 0.05$, * $p < 0.1$.

The results indicate that, in general, *Bsize* continues to exhibit a positive correlation with most earnings management proxies, primarily within the models examining firms from less corrupt economies. Columns 3 and 4 specifically show statistically significant coefficients for *Bsize*, regardless of the corruption level. These results confirm Kapoor and Goel (2017), Abdou et al. (2021), Githaiga et al. (2022), and Hasan et al. (2022) supporting the theory that the reason that the oversized board allows earnings management is the lack of efficient coordination and communication among the members. The figures concerning the variable Meet are weak, but a remarkable point is that in less corrupt countries, increases in the number of board meetings are associated with higher levels of accrual (column 2) and sales manipulation (column 8). The results confirm the findings of Hasan et al. (2022) only for low-corruption countries, supporting the notion that more active boards focus on other corporate issues, permitting managers to engage in opportunistic practices. In accordance with Table 3, the relationship between *abnDiscExp* and the independence ratio (*Indep*) is also significantly negative in both groups. On the contrary, although in Table 3, the association between *Indep* and *abnProd* is positive, in Table 4, the correlation in the low-corruption group is negative. In essence, in low-corruption countries, the increase in independent members improves the ability of boards to restrict real earnings management through production cost, verifying the outcomes of studies like Kang and Kim (2012) and Attia et al. (2022). The effect of board tenure on discretionary accruals is negative, but this relationship seems stronger in countries with higher corruption levels. In Table 4 the effect of *Tenure* on *abnProd* appears positive in low-corruption countries at a significant level. In other words, increases in board tenure contribute to a higher level of real earnings management though production cost, which is attributed to the fact that board member develop relationships with managers

relaxing the monitoring quality (Xie et al. 2003; Vafeas 2005). In line with Table 3, Table 4 provides statistically insignificant results for the variable Skills. According to Table 3, the condition in which the CEO also serves as chairman constrains the sales manipulation practices, but Table 4 confirms this relationship only for the group of observations of low-corruption environments. This finding is in line with two theories. The first concerns the concentrated ownership that prevails in European firms. CEO-chairmen, representing the major stockholders, pay intense attention on the financial reporting quality to safeguard the firm's value and stockholders' interests (Gavana et al. 2022; Attia et al. 2022). Moreover, when the CEO serves as chairman, more responsibilities are gathered in one person; hence, these CEOs avoid engaging in earnings management to preserve their reputation and their career (Badolato et al. 2014; Livnat et al. 2021).

Overall, this corruption-oriented analysis indicates that in some cases, the corruption level does not influence the effect of board efficiency on earnings management. For instance, the effect of board size is shown to be unaffected by the sample separation, as well as the effect of board independence on the manipulations through discretionary expenses. However, the board independence in the main findings is presented to lead to a higher level of earnings management, but in the additional analysis, the results suggest that in less corrupt countries, the board independence enhances the quality and efficacy of the board. In other cases, such as the effect of the annual board meetings (on accrual manipulation and sales manipulation), the second-stage analysis sheds light on which conditions confirm the results of the main analysis. For example, only in less corrupt environments are board meetings, tenure, and CEO duality significantly related to earnings management proxies.

### 4.2. Robustness Tests

This study incorporates two robustness tests to enhance the validity of the main research findings. Similar to Doukakis (2014) and Jiang et al. (2018), the first test involves implementing the proposed methodology on a limited sample, excluding firms from the United Kingdom, Germany, and Switzerland. It is notable that these three countries significantly influence the sample, accounting for 43.71% of the total dataset. Consequently, this robustness check aims to re-evaluate the model, with the primary objective of determining whether the findings are primarily driven by the data from these specific economic contexts. The second test includes the implementation of the bootstrap method. The bootstrap technique involves conducting multiple regressions by resampling observations from the data in memory several times (Christopoulos et al. 2019). In this case, the resampling is performed with replacement, and the replication number used for the bootstrap was 200.

### 4.2.1. Excluding Firms from the United Kingdom, Germany, and Switzerland

The findings from the initial robustness check in Table 5 are comparable to the main results presented in Table 3, although there are minor variations. Specifically, according to the Table 5, when considering the variable *Bsize*, the coefficients associated with *DACC*, *abnDiscExp*, and *abnProd* remain positive. However, in the model where *DACC* is the dependent variable, the significance appears weaker (from 0.05 to 0.1). Conversely, the significance is more robust in the model that utilizes abnProd as the dependent variable (from 0.1 to 0.05). Regarding the variable Meet, the results align with Table 3 regarding significance and the coefficient sign, except for the model where *DACC* is the dependent variable, where the coefficient is insignificant. In the case of the *Indep* variable, the coefficient improves in the model with *DACC* (from 0.1 to 0.05) and *abnCFO* (from no significance to 0.05) as dependent variables. The coefficient of the Tenure variable becomes more significant in the model where *abnProd* is the dependent variable. As for the independent variable, *Skills* has no statistical significance in its relationship with *DACC*, whereas the relationship with *abnProd* appears stronger (from no significance to 0.05). The impact of *Dual* on *abnDiscExp* becomes weaker, while the coefficient with *abnCFO* is now significant at a 0.05 level.

**Table 5.** Robustness check in reduced sample excluding countries.

|  | (1) | (2) | (3) | (4) |
|---|---|---|---|---|
|  | DACC | abnDiscExp | abnProd | abnCFO |
| Bsize | 0.017 * | 0.086 *** | 0.089 ** | −0.005 |
|  | (0.009) | (0.017) | (0.035) | (0.019) |
| Meet | 0.001 | −0.023 ** | −0.014 | 0.007 |
|  | (0.007) | (0.01) | (0.018) | (0.014) |
| Indep | 0 ** | −0.001 *** | −0.001 *** | 0 ** |
|  | (0) | (0) | (0) | (0) |
| Tenure | −0.014 ** | −0.05 *** | 0.044 ** | 0.002 |
|  | (0.006) | (0.01) | (0.02) | (0.006) |
| Skills | −0.001 | −0.003 | −0.032 ** | −0.005 |
|  | (0.006) | (0.006) | (0.014) | (0.01) |
| Dual | 0.006 | 0.041 *** | −0.002 | −0.01 ** |
|  | (0.005) | (0.008) | (0.021) | (0.005) |
| BIG4 | −0.009 * | 0.012 * | 0.023 * | −0.014 ** |
|  | (0.006) | (0.006) | (0.013) | (0.007) |
| ROA | 0.054 | −0.176 *** | 0.114 | 0.021 |
|  | (0.118) | (0.053) | (0.079) | (0.048) |
| Fsize | −0.01 *** | −0.024 *** | −0.034 *** | 0 |
|  | (0.003) | (0.003) | (0.006) | (0.003) |
| LEV | 0.003 | −0.101 *** | 0.04 | −0.071 ** |
|  | (0.018) | (0.02) | (0.044) | (0.032) |
| _cons | 0.187 *** | 0.523 *** | 0.59 *** | 0.029 |
|  | (0.054) | (0.071) | (0.132) | (0.055) |
| Observations | 712 | 1060 | 887 | 1069 |
| R-squared | 0.205 | 0.313 | 0.188 | 0.176 |

Robust standard errors are in parentheses. *** $p < 0.01$, ** $p < 0.05$, * $p < 0.1$.

Similarly, the robustness test approach is used in the separated sample according to the corruption level. According to Table 6 *Bsize*, the coefficients follow the signs of Table 4. The significance is inadequate when the dependent variable is *abnProd* in high-corruption countries and *abnCFO* in low-corruption countries. The coefficients of the variable Meet are significantly weaker since no statistical significance exists in any of the alternative models. The results for the variable *Indep* are similar to Table 4, while in some cases, the significance appears slightly improved (Columns 2 and 5). With regard to the variable Tenure, the sign of the coefficients is in line with Table 4, with some discrepancies. Although in Table 4, the effect of Tenure is significantly negative at a level of 1%, in Table 6, the significance is limited to 10%. However, when *abnDiscExp* is the dependent variable, the coefficients are improved regardless of the corruption level. The results for the variable *Skills* are similar to Table 4. The effects of the variable *Skills* on *abnDisc* in highly corrupt countries (Column 3) and the effect on *abnProd* in less corrupt countries (Collum 6) are enhanced. Concerning the variable Dual, the main difference in contrast to Table 4 is that the effect of *Dual* on *abnDiscExp* in both corruption levels appears significantly positive, while in Table 4, the results are too weak.

In aggregate, limiting the sample, the results are generally in line with the main research results when the proposed model is applied in the smaller sample without taking into consideration the corruption level. However, the results appear weaker when the sample is separated according to the level of corruption, since there are some discrepancies in the signs and the level of significance.

**Table 6.** Robustness check in reduced sample excluding countries and after separating the sample according to the corruption level.

| Dependent Variable: | DACC | | abnDiscExp | | abnProd | | abnCFO | |
|---|---|---|---|---|---|---|---|---|
| | (1) High Corruption | (2) Low Corruption | (3) High Corruption | (4) Low Corruption | (5) High Corruption | (6) Low Corruption | (7) High Corruption | (8) Low Corruption |
| Bsize | 0.021 * | 0.031 ** | 0.088 *** | 0.065 ** | 0.041 | 0.111 * | −0.034 | 0.023 |
| | (0.012) | (0.015) | (0.024) | (0.029) | (0.041) | (0.062) | (0.036) | (0.015) |
| Meet | 0.007 | 0 | −0.007 | 0.005 | 0.04 | −0.04 | 0.019 | 0.003 |
| | (0.008) | (0.015) | (0.015) | (0.017) | (0.025) | (0.031) | (0.026) | (0.009) |
| Indep | 0 * | 0 * | −0.0 *** | −0.002 *** | −0.001 * | −0.001 ** | 0 | 0 |
| | (0) | (0) | (0) | (0) | (0) | (0) | (0) | (0) |
| Tenure | −0.014 * | −0.007 | −0.072 *** | −0.039 ** | 0.009 | 0.04 | 0.011 | 0.003 |
| | (0.008) | (0.009) | (0.017) | (0.017) | (0.027) | (0.036) | (0.013) | (0.009) |
| Skills | 0.009 | −0.004 | 0.019 ** | −0.007 | 0.001 | −0.047 ** | −0.027 | 0.008 * |
| | (0.006) | (0.008) | (0.009) | (0.009) | (0.023) | (0.02) | (0.024) | (0.004) |
| Dual | 0.002 | 0.009 | 0.052 *** | 0.029 * | 0.002 | 0.057 * | 0.003 | −0.003 ** |
| | (0.005) | (0.011) | (0.01) | (0.018) | (0.03) | (0.03) | (0.008) | (0.006) |
| BIG4 | −0.004 | −0.017 | −0.003 | 0.028 ** | 0.02 | 0.021 | −0.011 | −0.017 ** |
| | (0.005) | (0.01) | (0.008) | (0.011) | (0.016) | (0.021) | (0.008) | (0.008) |
| ROA | −0.131 | 0.109 | −0.125 * | −0.149 ** | −0.049 | 0.013 | −0.055 | 0.095 |
| | (0.176) | (0.148) | (0.075) | (0.066) | (0.155) | (0.103) | (0.075) | (0.058) |
| Fsize | −0.01 *** | −0.015 *** | −0.025 *** | −0.031 *** | −0.045 *** | −0.013 | −0.001 | −0.006 * |
| | (0.004) | (0.004) | (0.005) | (0.005) | (0.008) | (0.009) | (0.004) | (0.003) |
| LEV | 0 | −0.02 | −0.125 *** | −0.069 ** | −0.026 | 0.134 * | −0.081 | −0.026 |
| | (0.024) | (0.041) | (0.031) | (0.034) | (0.059) | (0.078) | (0.05) | (0.028) |
| _cons | 0.174 *** | 0.385 *** | 0.538 *** | 0.851 *** | 0.853 *** | 0.467 ** | 0.177 ** | 0.151 ** |
| | (0.05) | (0.095) | (0.094) | (0.116) | (0.195) | (0.194) | (0.075) | (0.06) |
| Observations | 349 | 363 | 523 | 537 | 435 | 452 | 527 | 542 |
| R-squared | 0.206 | 0.21 | 0.271 | 0.218 | 0.181 | 0.104 | 0.112 | 0.175 |

Robust standard errors are in parentheses. *** $p < 0.01$, ** $p < 0.05$, * $p < 0.1$.

### 4.2.2. Bootstrap

The outcomes remain almost unchanged by employing the bootstrap method (Table 7), demonstrating a high level of consistency, albeit with minimal nuanced variations. Specifically, the variable relationship between the variable Meet with discretionary accruals (*DACC*) in the robustness check presents statistical significance at a level of 5%, while the main results of Table 3 are lower at a level of 10%.

**Table 7.** Robustness check with the resampling method of bootstrap.

| | (1) DACC | (2) abnDiscExp | (3) abnProd | (4) abnCFO |
|---|---|---|---|---|
| Bsize | 0.020 *** | 0.044 *** | 0.046 * | 0.014 |
| | (0.008) | (0.011) | (0.024) | (0.011) |
| Meet | 0.01 ** | −0.014 ** | −0.009 | 0.011 |
| | (00.05) | (0.006) | (0.016) | (0.007) |
| Indep | 0 * | −0.001 *** | 0 ** | 0 |
| | (0) | (0) | (0) | (0) |
| Tenure | −0.014 *** | −0.017 ** | 0.031 * | −0.002 |
| | (0.005) | (0.007) | (0.016) | (0.005) |
| Skills | 0.006 * | 0.005 | −0.008 | 0.003 |
| | (0.006) | (0.004) | (0.01) | (0.007) |
| Dual | 0 | 0.011 ** | −0.024 * | −0.013 *** |
| | (0.005) | (0.005) | (0.014) | (0.005) |
| BIG4 | −0.001 | 0.02 *** | 0.032 *** | −0.005 |
| | (0.005) | (0.006) | (0.011) | (0.005) |
| ROA | 0.374 *** | −0.094 *** | 0.449 *** | 0.572 *** |
| | (0.092) | (0.015) | (0.068) | (0.082) |
| Fsize | −0.009 *** | −0.015 *** | −0.023 *** | −0.006 *** |
| | (0.003) | (0.002) | (0.004) | (0.003) |
| LEV | 0.03 * | −0.081 *** | 0.02 | 0.016 |
| | (0.016) | (0.015) | (0.035) | (0.021) |

Robust standard errors are in parentheses. *** $p < 0.01$, ** $p < 0.05$, * $p < 0.1$.

Subsequently, the resampling approach of bootstrap is applied on the two groups of observations according to the corruption level. Table 8 presents some discrepancies. The significance of Tenure on *DACC*, in both groups, is presented weaker. Moreover, the coefficient of the variable *Meet* on *abnCFO* in low-corruption countries exhibits a significance of 10% in comparison to Table 4, which was more significant at a level of 5%.

**Table 8.** Robustness check with the resampling method of bootstrap in separating sample according to the corruption level.

| Dependent Variable: | DACC | | abnDiscExp | | abnProd | | abnCFO | |
|---|---|---|---|---|---|---|---|---|
| | (1) High Corruption | (2) Low Corruption | (3) High Corruption | (4) Low Corruption | (5) High Corruption | (6) Low Corruption | (7) High Corruption | (8) Low Corruption |
| Bsize | 0.009 | 0.029 *** | 0.04 *** | 0.042 *** | 0.086 *** | 0.033 | −0.004 | 0.027 ** |
| | (0.01) | (0.011) | (0.013) | (0.015) | (0.035) | (0.033) | (0.022) | (0.011) |
| Meet | 0 | 0.019 *** | −0.009 | −0.003 | −0.017 | 0.006 | 0.009 | 0.014 * |
| | (0.007) | (0.007) | (0.007) | (0.008) | (0.023) | (0.023) | (0.016) | (0.007) |
| Indep | 0 | 0 | −0.001 *** | −0.001 *** | 0 | −0.001 ** | 0 | 0 |
| | (0) | (0) | (0) | (0) | (0) | (0) | (0) | (0) |
| Tenure | −0.018 ** | −0.01 | −0.003 | −0.008 | 0.01 | 0.05 ** | 0.006 | −0.008 |
| | (0.008) | (0.007) | (0.008) | (0.01) | (0.026) | (0.024) | (0.008) | (0.006) |
| Skills | 0.007 | 0.004 | 0.008 * | 0.003 | −0.002 | −0.014 | −0.009 | 0.008 * |
| | (0.005) | (0.004) | (0.005) | (0.005) | (0.016) | (0.011) | (0.015) | (0.004) |
| Dual | 0 | −0.002 | −0.002 | 0.005 | −0.027 | −0.007 | −0.009 | −0.013 ** |
| | (0.006) | (0.007) | (0.007) | (0.008) | (0.026) | (0.016) | (0.007) | (0.005) |
| BIG4 | −0.005 | 0.001 | 0.028 *** | 0.027 *** | 0.053 *** | 0.017 | −0.01 | 0.001 |
| | (0.005) | (0.007) | (0.006) | (0.007) | (0.017) | (0.016) | (0.008) | (0.005) |
| ROA | 0.339 * | 0.394 *** | −0.081 *** | −0.091 *** | 0.416 * | 0.457 *** | 0.636 *** | 0.537 *** |
| | (0.213) | (0.103) | (0.014) | (0.019) | (0.243) | (0.104) | (0.182) | (0.121) |
| Fsize | −0.005 | −0.012 *** | −0.015 *** | −0.015 *** | −0.05 *** | −0.011 * | −0.004 | −0.008 *** |
| | (0.003) | (0.004) | (0.002) | (0.003) | (0.008) | (0.006) | (0.004) | (0.003) |
| LEV | 0.017 | 0.044 * | −0.063 *** | −0.042 ** | 0.022 | 0.024 | −0.019 | 0.065 ** |
| | (0.026) | (0.022) | (0.019) | (0.020) | (0.047) | (0.053) | (0.042) | (0.026) |
| _cons | 0.09 * | 0.186 ** | 0.321 *** | 0.369 *** | 0.906 *** | 0.394 *** | 0.074 | 0.083 |
| | (0.05) | (0.08) | (0.048) | (0.059) | (0.155) | (0.117) | (0.065) | (0.057) |
| Observations | 446 | 794 | 1347 | 1033 | 626 | 922 | 808 | 1050 |
| R-squared | 0.598 | 0.498 | 0.208 | 0.215 | 0.324 | 0.201 | 0.533 | 0.599 |

Robust standard errors are in parentheses. *** $p < 0.01$, ** $p < 0.05$, * $p < 0.1$.

## 5. Concluding Remarks

Many studies have investigated how board attributes influence the quality of financial information, providing diverse outcomes. However, the existing literature has paid little attention to international samples, particularly in the European area. This study addresses this gap by analyzing a dataset of 469 European companies to identify whether the firms of a diversified economic environment follow the behavior of the past literature that has focused mainly on specific economies. It aims to uncover the influence of the board of directors' characteristics on accrual and real earnings management.

The results reveal that, in less corrupt economies, an oversized board of directors lacks coordination and communication quality, facilitating earnings management through accruals and manipulations on discretionary expenses and sales, while in countries with higher corruption, increases in the board size lead to a higher level of manipulation through production cost and discretionary expenses. Moreover, while the past literature has provided evidence that the controlling ability of a board depends on the number of meetings and the degree of board independence, the results partially verify this assumption, documenting a negative impact of board meetings and the portion of independent members only on the discretionary expenses. On the contrary, in conditions of low corruption, the number of

meetings is positively associated with discretionary accruals and sales manipulation. Board independence mitigates earnings management through discretionary expenses regardless of the corruption level, while under low corruption, the ratio of independent members limits the level of manipulation through production cost. Another constraining determinant of earnings management is board tenure, which represents the experience and perceived quality of board members. The results demonstrated that the board tenure negatively affects the intensity of accruals and real earnings management through discretionary expenses. However, examining the group of low-corruption countries, the board tenure tends to induce increases in earnings management through production cost. Also, the condition when the CEO serves simultaneously as a chairman of the board facilitates managers to manipulate discretionary expenses while leading to lower levels of real earnings management through sales and production costs.

This research highlights the multidimensional nature of corporate governance's impact on financial information quality. The findings imply that the level of corruption is a determinant that enhances or mitigates the effect of board characteristics on the financial reporting quality but concerning specific board characteristics. The knowledge gained from this study has implications for theory and practice, offering valuable guidance for companies, regulators, and researchers dealing with enhancing transparency and accountability in financial reporting across European businesses. Firstly, the research outcomes offer valuable insights to stockholders concerning the efficiency of the boards of directors. According to the results, the board characteristics exhibit diverse effects across the different earnings management methods. For instance, board meetings and independence facilitate accrual earnings management but mitigate real earnings management. Secondly, the revealed results imply that investors need to consider corruption levels to comprehend whether the board characteristics contribute to improving financial reporting quality.

The limitation of this study concerns the lack of variables representing the personal characteristics of the board members, such as their age, educational background, relationship with the principal stockholders, and gender. The enhancement of this study with such variables could be an interesting expansion of the demonstrated findings. Also, future research could expand this study by employing worldwide datasets focusing on the differences between developed and emerging economies.

**Funding:** This research received no external funding.

**Data Availability Statement:** The employed data of this study are available from the corresponding author upon reasonable request.

**Conflicts of Interest:** The author declares no conflict of interest.

## Appendix A. Definition of Variables

| Variable | Definition |
| --- | --- |
| \|EM\| | Absolute value of discretionary accruals |
| abnCFO | Abnormal level of cash flows from operations |
| abnDiscExp | Abnormal level discretionary expenses |
| abnProd | Abnormal level of production cost |
| $A_{t-1}$ | Lagged total assets |
| BIG4 | Dummy variable which equals one for year-observations that the auditor is one of the four more prominent auditing firms, otherwise equals zero |
| Bsize | Natural logarithm of the board members |
| CFO | Cash flow from operations |
| DACC | Discretionary accruals |
| DISCEXP | Discretionary expenses |

| Variable | Definition |
|---|---|
| Dual | Dummy variable which equals one for firm years that the CEO serves simultaneously as chair of the board, otherwise equals zero |
| Fsize | Firm size-natural logarithm of total assets |
| Indep | The portion of independent board members divided by the total number of board members |
| LEV | Total debt divided by total assets |
| Meet | Natural logarithm of the number of the annual board meetings |
| PPE | Property plant and equipment |
| PROD | Production cost |
| ROA | Return on assets |
| S | Sales |
| Skills | The portion of the board members that have acquired educational financial or industry related skills |
| Tenure | Natural logarithm of average tenure (in years) of board members |
| TA | Total accruals |
| Δsales | Change of sales |

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
