# Peer review of "Earnings Management and Status of Corporate Governance under Different Levels of Corruption—An Empirical Analysis in European Countries"

_jrfm, doi:10.3390/jrfm16100458_

Round 1

Reviewer 1 Report

Dear author,

Thank you for sending your paper to the journal; the paper is interesting but needs further efforts in the following sections:

1-The employed data ended in 2019. It is recommended to extend up to 2022

2- Why has the paper concentrated on commercial corporations, whereas the production industries are desirable to conduct a study?

3- The most recent studies should enhance the theoretical issues and literature; some are listed here:

-Financial Ratios, Corporate Governance, and Macroeconomic Indicators in Predicting Financial Distress

-The relationship between economic complexity and green economy with earnings management

-The relationship between corporate governance and cost of equity: evidence from the ISIS era in Iraq

-The Effect of Corporate Governance Structure on Fraud and Money Laundering

-Nexus between corporate governance and earnings management in family and non-family firms

4-At the end of the conclusion, the implications are stated; however, at first, the mentioned implications are very general and second, they should evolve from the results of the study.

Author Response

Response to Reviewers

I would like to thank you the anonymous reviewers for the helpful comments during the reviewing process of my manuscript. Below, you will find the comprehensive responses along with the highlighted (in bold) changes made in the resubmitted files.

Response to Reviewer 1

Comment 1: The employed data ended in 2019. It is recommended to extend up to 2022

Response: The sample observations range from 2011 to 2019 to avoid being influenced by the effect of the Covid-19 pandemic. The level and the direction of earnings management can be affected by the economic cycles, especially under unexpected situations. In subsection 3.1, a highlighted (with bold) statement has been introduced to clarify the explanation for selecting this specific examined period. (Lines 328-331)

Comment 2: Why has the paper concentrated on commercial corporations, whereas the production industries are desirable to conduct a study?

Response: As mentioned in section 3.1, the employed sample includes commercial and industrial companies. (Lines 331-332)

Comment 3: The most recent studies should enhance the theoretical issues and literature; some are listed here

Response: Indeed, some of your recommended studies are closely related to this topic. The following references have been added in the text with bold.

  • “Mousavi, M., Zimon, G., Salehi, M. and StÄ™pnicka, N., 2022. The effect of corporate governance structure on fraud and money laundering. Risks, 10(9), p.176.” (Section 2, paragraph related to the impact of industrial and financial expertise on financial reporting quality, Lines 268 & 271)
  • “Salehi, M., Moradi, M. and Faysal, S., 2023. The relationship between corporate governance and cost of equity: Evidence from the ISIS era in Iraq. International Journal of Emerging Markets.” (Section 2, 3rd paragraph, Line 186)
  • “Ahmadi, Z., Salehi, M. and Rahmani, M., 2023. The relationship between economic complexity and green economy with earnings management. Journal of Facilities Management.” (Section 2, last paragraph, Line 316)

Comment 4: At the end of the conclusion, the implications are stated; however, at first, the mentioned implications are very general and second, they should evolve from the results of the study.

Response: The concluding remarks have been enriched with more elaborated implications, limitations and future research avenues (Lines 734-747).

Reviewer 2 Report

Dear Authors,

Congratulations on your theme and effort.

I trust that the suggestions provided in the attached document are valuable. My recommendation is to consider revising the article to address those points, as I believe it would improve the overall quality of the paper.

Kind regards,

Author Response

Response to Reviewers

I would like to thank you the anonymous reviewers for the helpful comments during the reviewing process of my manuscript. Below, you will find the comprehensive responses along with the highlighted (in bold) changes made in the resubmitted files.

 Title of the article

Comment 1:  “I found the term "European Area" potentially misleading, as it could be confused with the Euro Area. Perhaps "An Empirical Analysis in European Countries" would be a more suitable choice”.

Response: Thank you for the comment. The title of the article has been changes according to your recommendation.

 Section 1: Introduction

Comment 2: “I recommend including the concepts and definitions of "earnings management" in the introduction section to provide clarity for the reader. The definition presented in the Literature Review section is not very clear, and it would be beneficial to clarify the distinction between "real earnings management" and "accrual earnings management" for the reader's understanding”.

Response: Thank you for the comment. At the beginning of the second paragraph of the introduction, the definitions of earnings management, accrual, and real earnings management are presented (in bold letters). (Lines 41-48).

Comment 3: There are several references mentioned in the text that are not included in the bibliography section. Specifically, the references to Donatelli et al., 2014 (mentioned in lines 35 and 265), McNichols, 2002 (mentioned in lines 307 and 314), and Jamadar et al., 2021 (mentioned in line 476) are missing from the reference list. Please review the references and ensure that all cited sources are included in the bibliography section for proper citation and attribution.

Response: Thank you for the comment. Indeed, some references are not included in the bibliography section. The entire paper has been reviewed for other missing references.

Comment 4: “The text in lines 44-66 in the Introduction does not seem to fully reflect the mixed results mentioned in the Literature Review. To enhance the clarity and consistency of the paper, it would be advisable to ensure that the Introduction adequately introduces and aligns with the mixed results discussed in the Literature Review”.

Response: Thank you for the comment. The part of the text, that you mentioned, has been changed taking into consideration your comment (Highlighted with bold) (Lines 55-66 & 72-79)

Section 2. Literature Review

Comment 5: “Lines 149-58 - “(…) confirmed a positive association between board size and discretionary accruals (…)” & “On the contrary, other studies, (…) have demonstrated a negative correlation between large board size and accruals quality.” At a first glance, the statements do not appear contradictory. While the two statements describe different research findings (one focuses on the quantity (amount) of discretionary accruals, while the other focuses on the quality (reliability) of accruals), they are not necessarily contradictory. However, the authors' assertion of contradiction warrants additional clarification”.

Response: The second part of the text you mentioned has been rewritten in order to explain more clearly the difference between the two groups of studies (Lines 195-202). This part explains that  some studied provided evidence that the board size is positively related to the level of earnings management (discretionary accruals), while others documented that the board size mitigates the discretionary accruals.

Section 3. Steps of methodology – variables selection

3.2 Earnings management estimation

Comment 6: “The specification of Eq. (1) is not consistent with the specifications of Eq. (2)-(4), although in Eq. (1) “All variables are scaled by lagged total assets ….“ (lines 322-3). So all four models are standardized by the firm size, but only in Eq. (2)-(4) that standardization is made visible”.

Response: Thank you for the comment. Indeed, the terms of equation 1 have not been divided by lagged total assets. The equation has been reformed according to your comment.(Line 360).

Comment 7: “Lines 314-322 As it is common in the literature, please start by presenting the dependent variable (in this case, TA), followed by a brief description of the independent variables. Also, Dsales is in Eq. (1), but not explained; and DRev is explained but missing from the model”.

Response: Thank you for the comment. The paragraph related to the explanation of Eq. 1 has been rewritten to describe more clearly the dependent and independent variables (Bold letters) (Lines 364-373). The variable ΔREV is not missing from the model, but by mistake in the model is included as Δsales. The term is finally included as Δsales.

Comment 8: “In these four equations, it is not clear whether the authors are considering time-series models (as implied by the indexes of the variables) or panel data (i and t indexes)”

Response: Thank you for the comment. All models are implemented in panel data. Indexes i and t have been added in all models.

Comment 9: “Additionally, to enhance readability and clarity, italicizing the variable names (CFO; TA, Bsize, etc) helps distinguish them from the regular text and makes it easier for readers to identify the key terms in the statements. Please apply this suggestion consistently throughout the entire text”

Response: Thank you for the comment. Following your recommendation, the variable names have been written in italics to enhance readability and clarity.

3.3 Selected variables

Comment 10: “I noticed similarities between this section and the Literature Review. To enhance the paper, it could benefit from summarizing the variables in a table, similar to “Appendix A. variable descriptions” in Roychowdhury. S. (2006). Earnings management through real activities manipulation, Journal of Accounting and Economics, 42 (3) or “Panel A and B - Table 1” in Òªolak, Gönül and Öztekin, Özde (2021). The impact of COVID-19 pandemic on bank lending around the world, Journal of Banking & Finance, 133. The latter article also provided descriptive statistics for the primary variables, a common practice in the literature, though not mandatory. Only the control variables, in particular, require clarification”.

Response: Thank you for the comment. Appendix A has been added after the concluding remarks, presenting the definitions of the variables of all models and section 3.3. has been withdrawn. Also, the paragraph concerning the other control variables has been incorporated in the section “The empirical approach” after the variables’ presentation (Lines 436-456).

3.3 The empirical approach

Comment 11: “The incorporation of financial controls (Lines 484-88) is highly relevant. Nonetheless, it appears that country-level controls are absent. This represents my primary concern regarding the paper's methodology”.

Response: Thank you for the comment. Regarding the concern raised about the absence of country-level controls, this issue has been addressed by incorporating country fixed effects in our panel regression model. The inclusion of country fixed effects allows controlling for unobserved country-specific factors that could potentially influence the level of earnings management. By employing this approach, it is achieved accounting for cross-country variations that may impact our dependent variables, thereby mitigating potential biases originating from country-level heterogeneity

Comment 12: “The country- and time-fixed effects seem to be missing from Eq. (5)”.

Response: Thank you for the comment.  Eq.5 has been modified by adding country-, industry- and time-fixed effects, as you noticed.

Comment 13: “Kindly apply italics to the variable names (DACC, abnDiscExp, Bsize, etc.) in the text for improved readability and clarity. This will also eliminate the need for quotation marks around the variable names”.

Response: Thank you for the comment. All variables in the entire text have been applied in italics as you recommended.

  1. Empirical results and discussion

4.1 Main results

Comment 14: “Lines 499-500 “Multicollinearity occurs when independent variables are highly correlated, leading to biased results.” I’m not sure what the authors mean by “biased results”. Multicollinearity itself does not lead to biased coefficients in a regression model. Instead, it affects the reliability and stability of the coefficient estimates”.

Response: Thank you for the comment. In the first paragraph of section 4.1, the sentence related to the consequences of multicollinearity has been changed (bold letters). Your suggestion describes more precisely the effects of the multicollinearity problem. (Lines 466-467)

Comment 15: “The two following statements seem contradictory, “Table 2 reveals that Bsize has a robust negative connection with all indicators of earnings manipulation.” (line 510) and “The findings of Table 3 suggest a strong positive relationship between board size (Bsize) and earnings management practices.” (line 522-3). I suggest that the authors reconcile these apparently conflicting statements”.

Response: Thank you for the comment. Indeed, the correlation matrix and the regression provide conflicting results for the association between board size and earnings management. In the paragraph discussing the regression results, a passage has been added to explain that the presence of the control variables may influence the effect of an independent variable on the dependent and that the correlation matrix (Bold letters). (Lines 492-497).

Comment 16: “In lines 524; 534; 589; 615 referring to the p-value seems redundant”.

Response: Thank you for the comment. The parentheses with p-values have been deleted

Comment 17: “Please italicize the variable names (Bsize, etc) in the text, to enhance readability and clarity”.

Response: Thank you for the comment. All variables in the entire text have been applied in italics as you recommended.

4.2 Robustness tests

Comment 18: “Number of the subsection should be 4.2”.

Response: Thank you for the comment. The number of the section has been corrected

Comment 19: “The authors should provide clarification regarding the rationale behind excluding three specific countries for robustness testing. What criteria led to the selection of these particular countries?”

Response: Thank you for the comment. Regarding the first robustness test, the paragraph has been modified to clarify the reasons for excluding these three countries' observations (Bold letters, Lines 627-633) . These three countries represent approximately 43% of the entire sample, so it is interesting to identify whether these observations significantly influence the main research results.

Comment 20: “Furthermore, considering that the article's focus and novelty are related to corruption, it would be reasonable to expect Tables 5 and 6 to present results with the separated sample (low- and highcorruption, similarly to Table 4”.

Response: Thank you for the comment. Following your suggestion, the robustness tests have also applied in separated samples according to the corruption level. The relevant tables 6 & 8 and the discussion have been added (Lines 658-673, 677-681, 693-697).

Comment 21: “Please italicize all variable names (dependent and independent variables) in the text, to enhance readability and clarity”.

Response: Thank you for the comment. All variables in the entire text have been applied in italics as you recommended.

Concluding remarks

Comment 22: “Line 685 “However, the existing literature has paid little attention to international samples, particularly in the European Area.” The contribution to the existing literature, especially concerning international samples, particularly in the European Area, is not discussed in the introduction section”.

Response: Thank you for the comment. The fifth paragraph of the introduction has been added, discussing the need for international-focused research, providing arguments supporting that the European environment is appropriate for this study (Lines 101-115).

Reviewer 3 Report

w is that it is an interesting paper with a potential to contribute to the literature. It can, however, be improved further as follows:

1. Introduction: Please clarify your research questions, objectives, background motivation, theoretical and empirical motivation and the lines of contributions to the literature

2. Background – you need to explain why Europe is the appropriate context to conduct this study by exploiting regulatory, reform and policy issues and developments within the research context or setting.

3. Literature review: very recent papers directly related to your study are missing. I suggest to enrich your literature review and to add, among the others, the following studies:

Gavana, G., Gottardo, P., & Moisello, A. M. (2022). Related party transactions and earnings management in family firms: the moderating role of board characteristics. Journal of Family Business Management. 10.1108/JFBM-07-2022-0090

Viana, D. B. C., Lourenço, I. M. E. C., & Paulo, E. (2023). The effect of IFRS adoption on accrual-based and real earnings management: emerging markets' perspective. Journal of Accounting in Emerging Economies, 13(3), 485-508.

Le, Q. L., & Nguyen, H. A. (2023). The impact of board characteristics and ownership structure on earnings management: Evidence from a frontier market. Cogent Business & Management, 10(1), 2159748.

4. Empirical findings – please link your findings to the: (i) theory, (ii) empirics, (iii) context; and (iv) highlight their economic, academic/research and policy implications. Closely link up and cite the papers that you have discussed in the background, theory, literature review section to the findings you are presenting here.

5. Conclusion – Please outline a summary of findings, contributions, implications, limitations and avenues for future research.

Author Response

Response to Reviewers

I would like to thank you the anonymous reviewers for the helpful comments during the reviewing process of my manuscript. Below, you will find the comprehensive responses along with the highlighted (in bold) changes made in the resubmitted files.

Response to Reviewer 3

Comment 1: “Introduction: Please clarify your research questions, objectives, background motivation, theoretical and empirical motivation and the lines of contributions to the literature”.

Response: Thank you for the comment. Section 1 (Introduction) has been reformed according to your recommendations. Notably, the first five paragraphs summarize the existing literature, indicating that the prior studies have examined the impact of board characteristics on the quality of financial reporting, focusing primarily on accrual-based methods and employing samples of specific countries. These paragraphs present the motivations for focusing on accrual and real earnings management methods in an international sample. Also, the corruption level is introduced as a potential determinant of the impact of board characteristics on earnings management. The sixth paragraph presents the two primary research queries, and the seventh paragraph concerns the contributions of the research.

Comment 2: “Background – you need to explain why Europe is the appropriate context to conduct this study by exploiting regulatory, reform and policy issues and developments within the research context or setting”.

Response: Thank you for the comment. Following your suggestion, the fifth paragraph of the introduction has been added, discussing the need for international-focused research, providing arguments supporting that the European environment is appropriate for this study (Lines 101-115). .

Comment 3: “Literature review: very recent papers directly related to your study are missing. I suggest to enrich your literature review and to add, among the others, the following studies”:

  • Le, Q.L. and Nguyen, H.A., 2023. The impact of board characteristics and ownership structure on earnings management: Evidence from a frontier market. Cogent Business & Management, 10(1), p.2159748.
  • Gavana, G., Gottardo, P. and Moisello, A.M., 2022. Related party transactions and earnings management in family firms: the moderating role of board characteristics. Journal of Family Business Management.
  • Viana, D.B.C., Lourenço, I.M.E.C. and Paulo, E., 2023. The effect of IFRS adoption on accrual-based and real earnings management: emerging markets' perspective. Journal of Accounting in Emerging Economies, 13(3), pp.485-508.

Response: According to your recommendation, the following papers have been included in the text as references providing more recent relevant research findings

  • Le, Q.L. and Nguyen, H.A., 2023. The impact of board characteristics and ownership structure on earnings management: Evidence from a frontier market. Cogent Business & Management, 10(1), p.2159748. (Lines 274 & 297)
  • Gavana, G., Gottardo, P. and Moisello, A.M., 2022. Related party transactions and earnings management in family firms: the moderating role of board characteristics. Journal of Family Business Management.(Lines 79, 300 & 609)
  • Viana, D.B.C., Lourenço, I.M.E.C. and Paulo, E., 2023. The effect of IFRS adoption on accrual-based and real earnings management: emerging markets' perspective. Journal of Accounting in Emerging Economies, 13(3), pp.485-508. (Lines 93 & 176).
  •  

Comment 4: “Empirical findings – please link your findings to the: (i) theory, (ii) empirics, (iii) context; and (iv) highlight their economic, academic/research and policy implications. Closely link up and cite the papers that you have discussed in the background, theory, literature review section to the findings you are presenting here”.

Response: Thank you for the comment. Section 4.1, concerning the discussion of the generated findings, has been enhanced with additional explanations linking with theory and past literature.

Comment 5: “Conclusion – Please outline a summary of findings, contributions, implications, limitations and avenues for future research”.

Response: Thank you for the comment. The concluding remakes (Section5) have been enriched with more elaborated implications, limitations and future research avenues (Lines 734-747).

Round 2

Reviewer 1 Report

Dear author,

You improved the paper substantially and the current draft meets my academic expectations.

Reviewer 2 Report

Dear authors,

I highly appreciate your efforts in considering the reviewer’s suggestions. Congratulations! The paper has undergone significant improvements.

Kind regards,

Reviewer 3 Report

Congrats, the paper has been dramatically improved